# Silica-rich volcanism in the early solar system dated at 4.565 Ga

Poorna Srinivasan[1,2], Daniel R. Dunlap[3], Carl B. Agee[1,2], Meenakshi Wadhwa[3], Daniel Coleff[4], Karen Ziegler[1,2], Ryan Zeigler[5] & Francis M. McCubbin[5]

The ranges in chemical composition of ancient achondrite meteorites are key to understanding the diversity and geochemical evolution of planetary building blocks. These achondrites record the first episodes of volcanism and crust formation, the majority of which are basaltic. Here we report data on recently discovered volcanic meteorite Northwest Africa (NWA) 11119, which represents the first, and oldest, silica-rich (andesitic to dacitic) porphyritic extrusive crustal rock with an Al–Mg age of $4564.8 \pm 0.3$ Ma. This unique rock contains mm-sized vesicles/cavities and phenocrysts that are surrounded by quench melt. Additionally, it possesses the highest modal abundance (30 vol%) of free silica (i.e., tridymite) compared to all known meteorites. NWA 11119 substantially widens the range of volcanic rock compositions produced within the first 2.5–3.5 million years of Solar System history, and provides direct evidence that chemically evolved crustal rocks were forming on planetesimals prior to the assembly of the terrestrial planets.

[1] Institute of Meteoritics, University of New Mexico, Albuquerque, NM 87131, USA. [2] Department of Earth and Planetary Sciences, University of New Mexico, Albuquerque, NM 87131, USA. [3] Center for Meteorite Studies, School of Earth and Space Exploration, Arizona State University, Tempe, AZ 85287, USA. [4] Jacobs Technology, NASA Johnson Space Center, Mail Code XI3, 2101 NASA Parkway, Houston, TX 77058, USA. [5] NASA Johnson Space Center, Mail Code XI2, 2101 NASA Parkway, Houston, TX 77058, USA. Correspondence and requests for materials should be addressed to P.S. (email: psrinivasan@unm.edu)

Examination of ancient achondritic meteorites has provided evidence that differentiation was occurring on many planetesimals within the first 10 Ma of Solar System formation. Our record of differentiated meteorites from this time period mainly consists of rocks with relatively low silica content, similar to compositions observed in Earth's oceanic crust. However, newly discovered achondrites have recently provided evidence that crusts on planetesimals early in the Solar System were also comprised of more evolved rocks with higher silica contents than typical achondrites[1–3]. These evolved lavas, similar to Earth's average continental crust, were long thought to be exclusive to large, planet-sized bodies. Moreover, the origin of Earth's evolved crust arose through the presence of substantial water and plate tectonic processes that allow for tertiary crust formation[4], a process that is as of now, thought to be unique to Earth[5]. Although the presence of evolved granitic clasts in some lunar samples[6] and felsic impact spherules in some howardites[7] can be attributed to extensive differentiation of mafic parental magmas, it is not clear how well this process can explain the origin of all the evolved achondrite meteorites that have been discovered to date[1–3]. Furthermore, the geochemistry of these evolved achondrites are not consistent with tertiary crust formation, leaving open the possibility that additional mechanisms exist by which evolved melt compositions can occur in our Solar System. In the present study, we describe a newly discovered evolved achondrite Northwest Africa (NWA) 11119 that is unique among the existing evolved achondrite meteorites in its age, appearance, mineralogy, and bulk composition. NWA 11119 provides important constraints on the early Solar System processes that allow for the formation of evolved melt compositions.

The NWA 11119 meteorite was found in Mauritania in December 2016. The recovered specimen is a single stone of 453 g (Fig. 1), and resides at the Maine Mineral and Gem Museum. A 23 g subsample deposited at the Institute of Meteoritics (IOM) at the University of New Mexico in Albuquerque was used for this study. The specimen is highly friable and contains an unusual light-green fusion crust. Broken fragments of the interior show bright green and gray crystals up to 3 mm in size within a medium- to fine-grained matrix. Spherical vesicles and irregular-shaped cavities were observed throughout the stone.

## Results

**Petrology of NWA 11119.** Petrologic examination of NWA 11119 was conducted on the IOM deposit sample (Supplementary Fig. 1; $4.2 \times 3.3 \times 2.3$ cm) as well as a polished slice ($0.95 \times 0.55$ cm) of the deposit sample embedded in a 1-inch epoxy mount. A false-colored Si–Fe–Al X-ray mosaic of the slice is shown in Fig. 2. NWA 11119 exhibits a porphyritic texture composed of millimeter-sized phenocrysts of pyroxene (<3.7 mm), feldspar (<4 mm), and silica (<3.4 mm) surrounded by a medium- to fine-grained quenched groundmass comprised of crystals that range in size from submicrometer to hundreds of micrometers (<300 μm) (Supplementary Fig. 2). Modal abundances of matrix and phenocryst phases were calculated by X-ray computed tomography (XCT) and through modal recombination of electron microprobe and scanning electron microscope (SEM) data. The 3D results of the deposit sample from the XCT instrument were obtained through density contrasts and indicated that NWA 11119 comprises 87.8% phenocrysts, 11% groundmass, and 1.2% vesicles/cavities (Supplementary Movies 1 and 2). The phenocrysts comprise 56% feldspar, 30% silica, and 14% pyroxene.

**Phenocryst compositions.** Chemical compositions of the phenocrysts (Table 1) exhibit moderate–minor variations and are represented by the following: clinopyroxene (augite; $Fs_{6.0\pm2.1}$

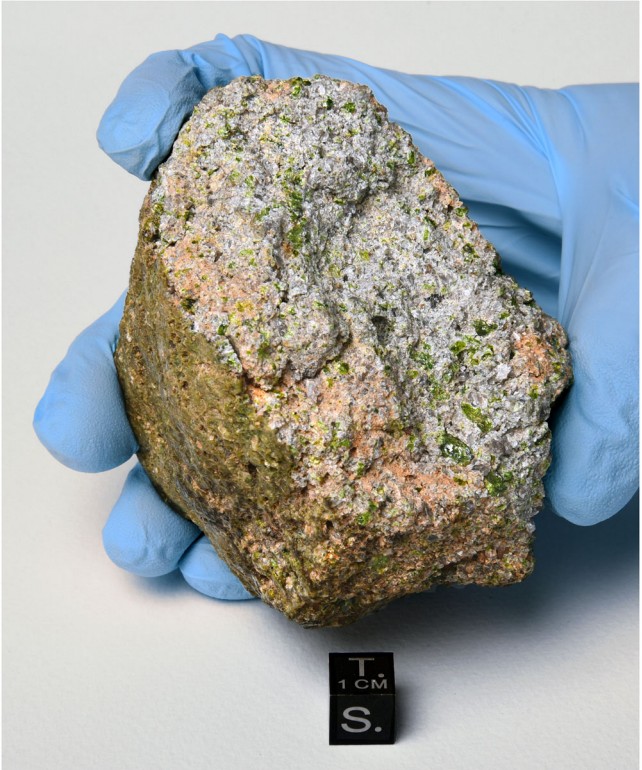

**Fig. 1** Photograph of the main mass of NWA 11119 that was found in Mauritania in December 2016. The rock has a light-green fusion crust, and the broken interior shows light-green and gray-colored crystals. Photo credit: © 2017 B. Barrett/Maine Mineral & Gem Museum

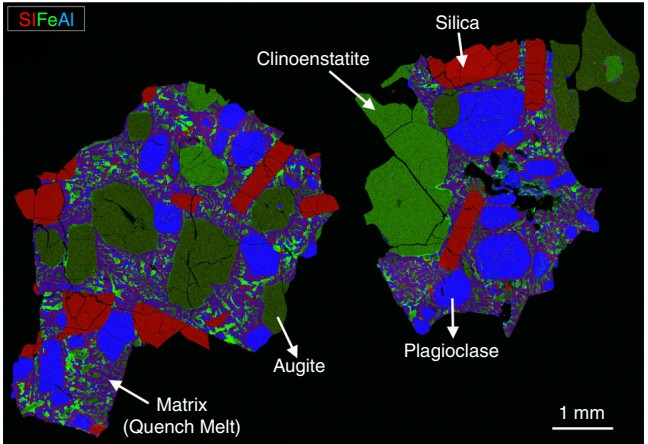

**Fig. 2** False-colored X-ray map of NWA 11119 showing mm-sized phenocrysts of silica, clinoenstatite, augite, and plagioclase. Zoning was not observed in silica or pyroxene phenocrysts, but normal zoning appears in plagioclase phenocrysts. The phenocrysts are surrounded by quenched groundmass consisting of highly zoned crystals that range in size from submicrometer to hundreds of micrometers

$Wo_{38.1\pm1.1}$; $Cr_2O_3 = 0.7$–$1.2$ wt.%; Mg# [$100 \cdot Mg/(Mg + Fe)$] = 90), orthopyroxene (enstatite; $Fs_{9.5\pm0.3}Wo_{4.6\pm0.1}$; $Cr_2O_3 = 0.8$–$0.9$ wt.%; Mg# = 90) (Supplementary Fig. 3 and Supplementary Data 1), plagioclase ($An_{87.3\pm7.2}Ab_{12.6\pm7.1}Or_{0.1\pm0.2}$) (Supplementary Fig. 4 and Supplementary Data 1), and silica (~97.2 wt.% $SiO_2$ and ~0.8 wt.% $Al_2O_3$) (Supplementary Data 1). The molar Fe/Mn ratio of pyroxene phenocrysts is 4.6–7.8 (Supplementary Fig. 5); with the exception of some winonaites, this range is lower than all

**Table 1 Mineral compositions (wt.%) for NWA 11119 determined by EPMA**

| | Augite | | | | | | | Enstatite | | | | | | |
|---|---|---|---|---|---|---|---|---|---|---|---|---|---|---|
| | All phenocrysts | | | Core[a] | | Rim[a] | | All phenocrysts | | | Core[a] | | Rim[a] | |
| | Average | Min | Max | Average | St. dev. | Average | St. dev. | Average | Min | Max | Average | St. dev. | Average | St. dev. |
| $SiO_2$ | 54.19 | 52.07 | 55.63 | 54.46 | 0.33 | 54.00 | 0.75 | 56.74 | 55.73 | 57.62 | 56.73 | 0.36 | 56.53 | 0.19 |
| MgO | 19.85 | 16.70 | 21.51 | 20.17 | 0.26 | 19.63 | 0.89 | 32.11 | 31.32 | 32.84 | 32.17 | 0.31 | 31.96 | 0.13 |
| $Na_2O$ | 0.08 | 0.04 | 0.15 | 0.08 | 0.01 | 0.09 | 0.02 | 0.01 | 0.00 | 0.03 | 0.01 | 0.01 | 0.01 | 0.01 |
| $Al_2O_3$ | 1.46 | 1.18 | 2.28 | 1.37 | 0.04 | 1.74 | 0.34 | 0.89 | 0.79 | 1.13 | 0.93 | 0.10 | 0.92 | 0.15 |
| $P_2O_5$ | 0.01 | 0.00 | 0.05 | 0.00 | 0.01 | 0.01 | 0.01 | 0.01 | 0.00 | 0.03 | 0.01 | 0.01 | 0.00 | 0.00 |
| CaO | 18.83 | 16.10 | 19.34 | 19.07 | 0.05 | 18.41 | 0.98 | 2.38 | 2.31 | 2.47 | 2.36 | 0.03 | 2.36 | 0.01 |
| $K_2O$ | 0.00 | 0.00 | 0.07 | 0.00 | 0.01 | 0.00 | 0.00 | 0.00 | 0.00 | 0.02 | 0.00 | 0.00 | 0.00 | 0.00 |
| $TiO_2$ | 0.36 | 0.23 | 1.08 | 0.31 | 0.01 | 0.48 | 0.22 | 0.16 | 0.13 | 0.19 | 0.16 | 0.02 | 0.15 | 0.00 |
| $Cr_2O_3$ | 0.79 | 0.69 | 1.18 | 0.74 | 0.01 | 0.92 | 0.15 | 0.81 | 0.76 | 0.85 | 0.82 | 0.03 | 0.80 | 0.01 |
| NiO | 0.01 | 0.00 | 0.04 | 0.01 | 0.01 | 0.01 | 0.01 | 0.01 | 0.00 | 0.04 | 0.01 | 0.01 | 0.01 | 0.01 |
| FeO | 3.79 | 3.25 | 9.63 | 3.36 | 0.03 | 4.31 | 0.98 | 6.34 | 6.22 | 7.61 | 6.33 | 0.04 | 6.33 | 0.05 |
| MnO | 0.72 | 0.60 | 1.47 | 0.66 | 0.02 | 0.81 | 0.14 | 0.94 | 0.90 | 0.99 | 0.94 | 0.02 | 0.94 | 0.01 |
| Total | 100.08 | 98.18 | 101.92 | 100.24 | 0.59 | 100.41 | 0.49 | 100.39 | 98.56 | 101.82 | 100.48 | 0.57 | 100.01 | 0.32 |

| | Plagioclase | | | | | | | Silica | | | | | | |
|---|---|---|---|---|---|---|---|---|---|---|---|---|---|---|
| | All phenocrysts | | | Core[a] | | Rim[a] | | All phenocrysts | | | Core[a] | | Rim[a] | |
| | Average | Min | Max | Average | St. dev. | Average | St. dev. | Average | Min | Max | Average | St. dev. | Average | St. dev. |
| $SiO_2$ | 47.22 | 44.69 | 53.10 | 46.31 | 0.09 | 50.86 | 2.10 | 97.36 | 96.63 | 98.35 | 97.00 | 0.12 | 97.57 | 0.54 |
| MgO | 0.32 | 0.11 | 0.49 | 0.34 | 0.01 | 0.28 | 0.12 | 0.00 | 0.00 | 0.01 | 0.00 | 0.00 | 0.00 | 0.00 |
| $Na_2O$ | 1.39 | 0.87 | 3.68 | 0.98 | 0.04 | 2.76 | 0.77 | 0.12 | 0.08 | 0.28 | 0.12 | 0.01 | 0.14 | 0.07 |
| $Al_2O_3$ | 34.52 | 30.14 | 37.12 | 35.21 | 0.32 | 31.87 | 1.64 | 0.74 | 0.61 | 1.44 | 0.71 | 0.01 | 0.81 | 0.29 |
| $P_2O_5$ | 0.01 | 0.00 | 0.04 | 0.01 | 0.01 | 0.01 | 0.01 | 0.00 | 0.00 | 0.03 | 0.00 | 0.01 | 0.01 | 0.01 |
| CaO | 17.46 | 13.11 | 18.61 | 18.18 | 0.08 | 14.81 | 1.55 | 0.09 | 0.04 | 0.29 | 0.09 | 0.01 | 0.11 | 0.08 |
| $K_2O$ | 0.02 | 0.00 | 0.20 | 0.01 | 0.00 | 0.09 | 0.06 | 0.14 | 0.01 | 0.17 | 0.14 | 0.00 | 0.13 | 0.06 |
| $TiO_2$ | 0.02 | 0.00 | 0.10 | 0.01 | 0.01 | 0.05 | 0.03 | 0.10 | 0.07 | 0.18 | 0.09 | 0.01 | 0.11 | 0.03 |
| $Cr_2O_3$ | 0.00 | 0.00 | 0.02 | 0.00 | 0.01 | 0.00 | 0.00 | 0.00 | 0.00 | 0.01 | 0.00 | 0.00 | 0.00 | 0.00 |
| NiO | 0.01 | 0.00 | 0.04 | 0.01 | 0.01 | 0.01 | 0.01 | 0.01 | 0.00 | 0.04 | 0.01 | 0.02 | 0.01 | 0.01 |
| FeO | 0.17 | 0.09 | 0.50 | 0.11 | 0.01 | 0.36 | 0.12 | 0.02 | 0.00 | 0.04 | 0.02 | 0.01 | 0.03 | 0.01 |
| MnO | 0.03 | 0.01 | 0.06 | 0.02 | 0.01 | 0.04 | 0.01 | 0.00 | 0.00 | 0.01 | 0.00 | 0.01 | 0.00 | 0.00 |
| Total | 101.17 | 99.71 | 101.94 | 101.19 | 0.31 | 101.13 | 0.26 | 98.58 | 98.04 | 99.78 | 98.19 | 0.11 | 98.92 | 0.59 |

[a]Core and rim values were obtained by averaging the first and last three analyses of each line array

achondritic groups[8]. This Fe/Mn ratio suggests formation in a reduced environment with a low oxygen fugacity below the iron-wüstite (IW) buffer and above the Mn–MnO buffer, similar to the reduced silicate mineralogy in winonaites[9]. The orthopyroxene and clinopyroxene phenocrysts exhibit an OPX–CPX Mg–Fe exchange $K_D$ ratio $[K_D = (X_{MgO}/X_{FeO})^{OPX}*(X_{FeO}/X_{MgO})^{CPX}]$ of $0.88 \pm 0.10$ (Supplementary Data 2 and Supplementary Fig. 6). This $K_D$ value is in the range of equilibrium orthopyroxene and clinopyroxene assemblages in terrestrial rocks such as pyroxene andesites ($K_D = 0.86$) and ultramafic inclusions in basaltic rocks ($K_D = 0.75$–$1.51$)[10]. Plagioclase phenocrysts exhibit normal zoning with rims that are more albitic than the cores. X-ray diffraction (XRD) patterns collected from a powdered fragment of NWA 11119 (Supplementary Fig. 7) revealed that the silica phase is tridymite. The tridymite phenocrysts appear either as long lathes or large anhedral grains. The pyroxene and plagioclase phenocrysts are subhedral to anhedral.

**Matrix material (quench melt).** The matrix consists of grains that are nearly up to ~300 mm in size, are highly zoned, and exhibit substantial variability among matrix grains of an individual phase (Supplementary Data 3). Submicrometer-to-micrometer-sized grains of ulvöspinel, ilmenite, troilite, tranquillityite, zircon, Fe metal, fayalite, and tsangpoite were also observed in the matrix. The texture of the matrix, coupled with the highly variable chemical compositions of matrix minerals, indicates rapid quenching from high temperatures. Consequently, we collected density-corrected quantitative X-ray maps of the groundmass to estimate the composition of the melt interstitial to

the phenocryst phases, and this melt composition is represented in Supplementary Data 4. The calculated matrix composition shows low total alkali abundances (<1.5 wt.%) and elevated silica content (~61.28 wt.%). In fact, the bulk composition of the matrix plots near the andesite–dacite boundary on the total alkali versus silica (TAS) diagram (Fig. 3a). To assess the relationship between the matrix and the phenocrysts, we evaluated the Mg–Fe exchange equilibria between pyroxenes and the groundmass. The orthopyroxene and clinopyroxene phenocrysts exhibit melt-pyroxene Mg–Fe exchange $K_D$ ratios $[K_D = (X_{MgO}/X_{FeO})^{Liquid}* (X_{FeO}/X_{MgO})^{pyroxene}]$ of $0.25 \pm 0.05$ and $0.22 \pm 0.06$, respectively (Supplementary Data 2). These values overlap with the range of equilibrium values for OPX-melt of 0.2–0.3[11,12] and CPX-melt of $0.27 \pm 0.03$[13], indicating that the phenocrysts could have been in equilibrium with the melt in which they were hosted prior to quench.

**Major and minor element bulk composition of NWA 11119.** The major element bulk composition of NWA 11119 was calculated using the modal abundances of each phase from the XCT data and the phase compositions determined by EPMA (Supplementary Data 4) following the procedures of ref. [14] and densities of ref. [15]. The calculated bulk rock composition indicates total alkali abundances (i.e., $Na_2O + K_2O$) of 0.93 wt.% and a silica content of 61.37 wt.% (Fig. 3a). If the bulk rock composition of NWA 11119 represents that of a melt composition, it is an andesite (Fig. 3a), however, we cannot rule out the possibility that NWA 11119 represents a partial cumulate rock. NWA 11119 has a Mg# of 84, considerably higher than terrestrial orogenic

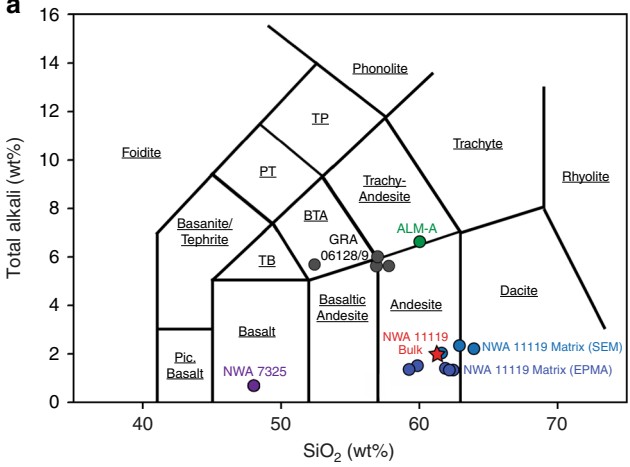

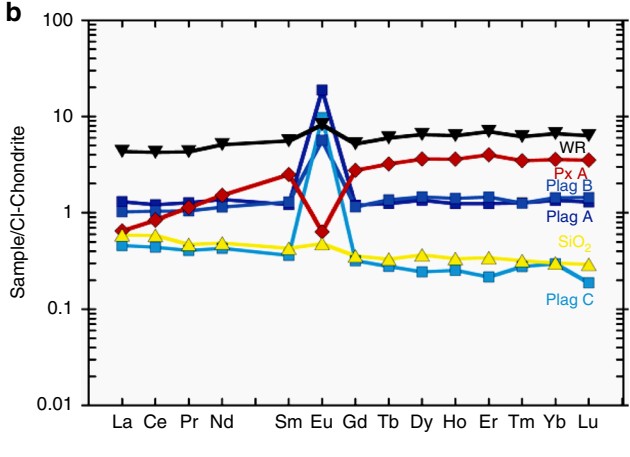

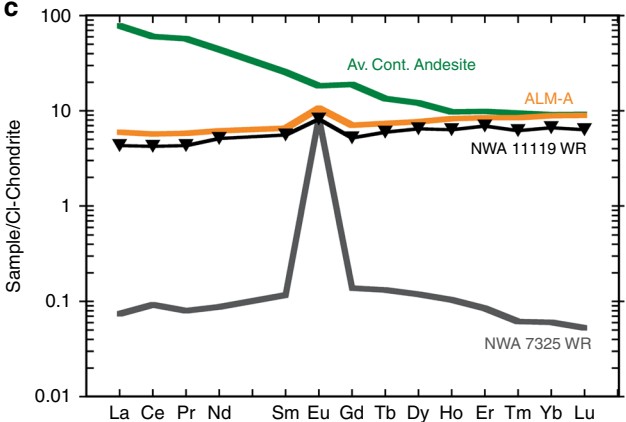

**Fig. 3** Chemical composition of NWA 11119. **a** Plot of total alkalis vs. silica content showing the compositions of the bulk and matrix of NWA 11119. The bulk rock has low alkali content, and measured values for the matrix show an evolved andesitic/dacitic composition. For comparison, bulk compositions for ALM-A and NWA 7325 are also shown. **b** CI-normalized rare earth element (REE) abundances in the NWA 11119 whole rock (WR) and mineral separates (Px = pyroxene; Plag = plagioclase; SiO₂ = Silica). **c** Comparison of CI-normalized REE abundances in whole rocks of NWA 11119 (black inverted diamonds), ALM-A (orange), NWA 7325 (gray), and average continental andesite (green). Compositional data for ALM-A from ref. [3], NWA 7325 from refs. [21,39], average continental andesite from ref.[54], and CI chondrites from ref. [55] Errors on our REE data (2 sigma) are typically ±10% and are smaller than the data points

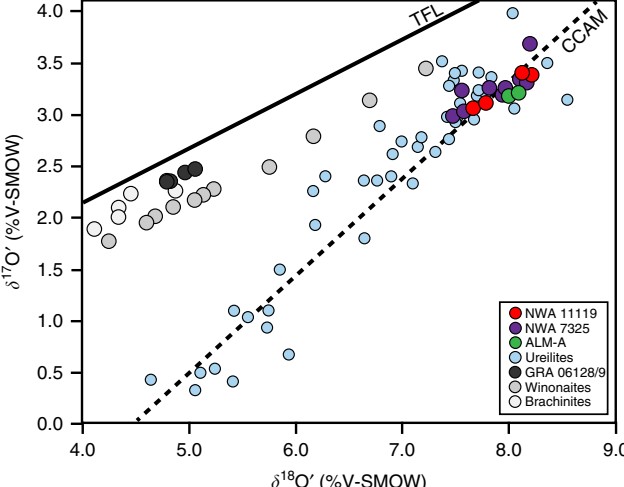

**Fig. 4** Oxygen three-isotope plot with various achondritic groups. NWA 11119 plots along the carbonaceous chondrite anhydrous minerals (CCAM) line with ureilites, the trachyandesitic fragment ALM-A from Almahata Sitta, and ungrouped achondrite NWA 7325. Literature data for all samples besides NWA 11119 are obtained from refs. [1,3,17,20]

andesites, which average Mg#'s of around 60[10]. The total alkali content of NWA 11119 is lower than typical terrestrial andesites and dacites, which have values >4 wt.%[16]. The silica content of the matrix is similar to the bulk rock.

**Rare earth element abundances of NWA 11119.** Rare earth element (REE) concentrations were determined in a whole-rock (WR) fraction and mineral separates (Px A, Plag A, Plag B, Plag C, and SiO₂) (Fig. 3b, c and Supplementary Data 5). Care was taken to ensure that the mineral separates (purified by hand picking under an optical microscope) were as pure as possible. Nevertheless, small amounts of other phases may be present in these separates as inclusions or minute-attached grains. As such, the REE patterns of the separates are characteristic of their dominant phases and most likely do not represent pure mineral compositions. Specifically, Px A, which shows light-REE depletion (CI-normalized La/Yb = 0.2) and a negative Eu anomaly (Eu/Eu* = 0.2, where Eu* is the value interpolated between CI-normalized abundances of Sm and Gd), is predominantly pyroxene. The Plag A, B, and C separates show relatively flat REE patterns (CI-normalized La/Yb = 0.7–1.6) with a pronounced positive Eu anomaly (Eu/Eu* = 4.6–28.6). While each of these three is dominated by plagioclase, Plag C is more so than the other two (given the significantly larger Eu anomaly). The tridymite-rich separate has a slight LREE enrichment (CI-normalized La/Yb = 2.0) with a small positive Eu anomaly (Eu/Eu* = 1.2). The NWA 11119 WR has a slight LREE-depletion (CI-normalized La/Yb = 0.7) with La ~4× CI chondrite, and a small positive Eu anomaly (Eu/Eu* = 1.5).

**Oxygen isotopic composition of NWA 11119.** Oxygen isotope ratios are used to distinguish between planetary bodies and/or planetary reservoirs. The NWA 11119 bulk sample has oxygen isotope ratios of δ18O = 8.226, 7.781, 8.121, 7.671, δ17O = 3.364, 3.133, 3.398, 3.046, and Δ17O = −0.979, −0.975, −0.890, and −1.004 per mil for four bulk rock fragments, respectively (Supplementary Data 6). NWA 11119 plots on the carbonaceous chondrite anhydrous mineral (CCAM) line on an oxygen three-isotope diagram (Fig. 4), which is uncommon for achondrites. Ureilites, which are ultramafic achondrites or fragmental/regolith breccias[17–19], the trachyandesitic clast ALM-A in the anomalous

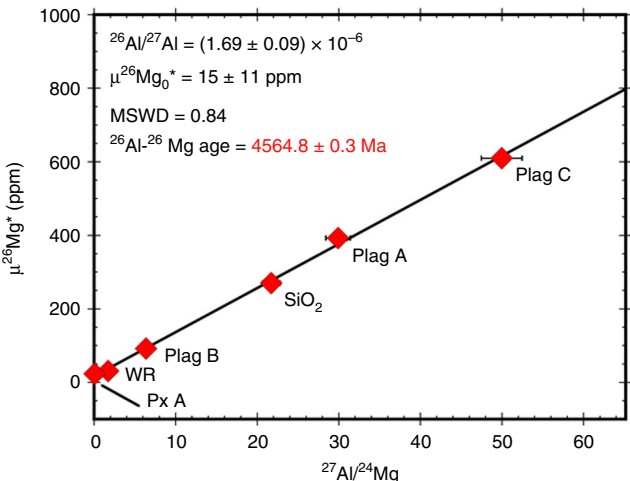

$^{26}Al/^{27}Al = (1.69 \pm 0.09) \times 10^{-6}$

$\mu^{26}Mg_0^* = 15 \pm 11$ ppm

MSWD = 0.84

$^{26}Al$-$^{26}Mg$ age = 4564.8 ± 0.3 Ma

**Fig. 5** The Al–Mg internal isochron defined by whole rock (WR) and mineral separates of NWA 11119. This isochron corresponds to a $^{26}Al/^{27}Al$ ratio of (1.69 ± 0.09) × 10$^{-6}$. Relative to the D'Orbigny age anchor, NWA 11119 has a $^{26}Al$-$^{26}Mg$ age of 4564.8 ± 0.3 Ma. The uncertainties on the μ$^{26}Mg^*$ values are either the internal 2SE errors (2× the standard error) on repeat analyses of the sample or the 2 SD (2× the standard deviation) external reproducibility based on repeat analyses of standards, whichever is larger; the uncertainties on the $^{27}Al/^{24}Mg$ ratios are ±5%

polymict ureilite Almahata Sitta[3], and the basaltic ungrouped achondrite NWA 7325[20–22] are the only other known achondrites that plot at high δ$^{18}$O values (>4‰) along the CCAM line. The ungrouped achondrites NWA 3133[23], NWA 8186[24,25], and NWA 7822[26] plot at lower δ$^{18}$O (<4‰) along the CCAM line, and show strong links to carbonaceous chondrite meteorite groups.

**Age constraints on NWA 11119.** The $^{26}Al$–$^{26}Mg$ isotope systematics were determined for whole rock (WR) and mineral separates of NWA 11119 (Supplementary Data 7). These define an isochron corresponding to a $^{26}Al/^{27}Al$ ratio at the time of crystallization of this sample of (1.69 ± 0.09) × 10$^{-6}$ (Fig. 5). Previously, the highest value reported for an achondrite internal $^{26}Al$–$^{26}Mg$ isochron was (1.28 ± 0.07) × 10$^{-6}$ (for the ancient cumulate eucrite Asuka 881394[27]). Relative to the D'Orbigny angrite age anchor[28,29], the $^{26}Al$–$^{26}Mg$ age of NWA 11119 is 4564.8 ± 0.3 Ma (Fig. 5). It is also possible to calculate a $^{26}Al$–$^{26}Mg$ age relative to the first-formed refractory solids in the solar protoplanetary disk that are characterized by a canonical $^{26}Al/^{27}Al$ ratio of 5.2 × 10$^{-5}$ [30]. However, the reported Pb–Pb ages for these refractory solids span ~0.6 Ma[30–32], leading to a range of $^{26}Al$–$^{26}Mg$ ages of 4563.7 ± 0.2 –4564.4 ± 0.3 Ma for NWA 11119. Consideration of the full range of these $^{26}Al$–$^{26}Mg$ ages (obtained relative to the D'Orbigny angrite or the refractory inclusions) and the Pb–Pb ages of the refractory inclusions[30–32] indicates that NWA 11119 was formed 2.5–3.5 Ma after the first-formed refractory solids in the solar protoplanetary disk.

## Discussion
Examples of ancient igneous activity on planetesimals within a few million years of Solar System formation (assumed to coincide with the formation of the first refractory solids in the solar protoplanetary disk[30–32]) are dominated by samples that exhibit basaltic–ultramafic bulk rock compositions. For example, internal Pb–Pb and $^{26}Al$–$^{26}Mg$ isochrons for both the angrite D'Orbigny, which contains vesicles and glass-filled cavities, and the cumulate eucrite Asuka 881394, which has a granulitic texture, indicate that basaltic crusts formed on their distinct differentiated parent

bodies well within ~5 Ma of Solar System formation[27,28,33,34]. Additionally, internal Pb–Pb and $^{26}Al$–$^{26}Mg$ isochrons for the ungrouped achondrite NWA 7325 provide evidence that basaltic crusts were forming on a reduced, differentiated parent body by ~4563 Ma[21,35,36]. Geochronologic data for the recently discovered volcanic and subvolcanic meteorites with rare trachyandesitic compositions are somewhat limited but provide some evidence that their parent bodies also differentiated early. The $^{26}Al$–$^{26}Mg$ model age for the ungrouped trachyandesitic paired achondrites GRA 06128/9[1,37,38] suggests that differentiation and plagioclase accumulation on the oxidized, volatile-rich, differentiated body may have occurred as early as ~4565 Ma[38]. Similarly, a $^{26}Al$–$^{26}Mg$ model age for the trachyandesitic fragment, ALM-A, from the anomalous polymict ureilite Almahata Sitta indicates that evolved crusts may have formed on a volatile-rich, differentiated body by ~4561 Ma[3]. The bulk composition of NWA 11119, along with its early formation age as indicated by its $^{26}Al$–$^{26}Mg$ internal isochron, demonstrates unambiguously that andesitic–dacitic volcanism occurred on a differentiated parent body within the first 2.5–3.5 Ma of Solar System history, extending both the age and degree of chemical evolution of planetary crustal materials on differentiated bodies in our Solar System.

NWA 11119 shares a number of textural and geochemical similarities with other ancient volcanic achondrites. For example, NWA 11119 has textural similarities to D'Orbigny and other angrites (e.g., vesicles), but it shares isotopic and petrologic links to the achondrites NWA 7325 and ALM-A in regard to oxygen isotopes (Fig. 4) and the presence of Cr-bearing high-Ca pyroxene[3,21]. NWA 11119 also shares similarities in whole-rock REE abundances with the ALM-A fragment (Fig. 3c). Nevertheless, NWA 11119 has other petrologic and geochemical distinctions (e.g., high free silica content and low total alkali content) that make it a unique crustal rock compared to these samples. Figure 3a shows the bulk rock content of NWA 11119 compared to the isotopically similar NWA 7325[39] and ALM-A[3] samples, and the trachyandesitic GRA 06128/9[1] samples. Although NWA 11119 shows notable geochemical similarities to NWA 7325 and ALM-A, it possesses a distinct bulk rock composition compared to these two samples. Furthermore, the bulk rock composition and pyroxene compositions of NWA 11119 have distinct Fe/Mn ratios compared to NWA 7325 and ALM-A (Supplementary Fig. 5), indicating that NWA 11119 is an andesitic achondrite that may be derived from a previously unsampled parent body.

The formation mechanisms of andesitic crustal materials are highly debated[4,40,41]. On Earth, ideas range from formation as primary melts from mantle material to secondary products of basaltic partial melts[40]. Plate tectonics and mantle volatile contents are also thought to play significant roles in the production of andesitic crusts[4,41]. The formation mechanisms of andesitic materials in asteroidal bodies are more uncertain, and it is still an area of intense research. Melting of chondritic precursors are typically implicated in the formation of achondrites, and the relatively unfractionated REE pattern of the NWA 11119 whole rock (Fig. 3c) is consistent with such a scenario. However, the petrogenetic origin of NWA 11119 cannot be easily linked to known melting processes of chondritic precursor materials. Nevertheless, several experimental studies on partial melting of chondritic materials have reported the formation of evolved melt compositions similar to the SiO$_2$ content of NWA 11119[42–45]. These studies may provide insights into the melting processes that occurred to produce NWA 11119.

Disequilibrium partial melting experiments conducted on an L6 ordinary chondrite produced a basaltic to basaltic–andesite silicate melt composition, but the melt contained significantly higher FeO content (~11–16 wt.%) and a range of total alkali contents (up to ~5 wt.%)[43]. Experiments on an H chondrite bulk

composition produced basaltic to dacitic melt compositions at IW-1, but the FeO abundances (9–14 wt.%) and alkali abundances (3.5–5 wt.%) of the melt were also too high[45]. Partial melting experiments on an R4 rumuruti chondrite produced a trachyandesitic silicate melt composition, but those experiments produced melts with even higher alkali abundances (~9 wt.%)[44]. Equilibrium partial melting experiments on an EH4 enstatite chondrite produced an andesitic–dacitic silicate melt with low alkali content[42]; however, FeO abundances (<0.6 wt.%) were well below those of NWA 11119, owing to the highly reduced nature of the enstatite chondrite parent body.

The EH4, H, and R4 experimental studies highlight the two mechanisms for producing silica-rich melts through equilibrium partial melting of chondritic precursor materials: (1) reducing to highly reducing conditions (akin to the H and EH4 studies) promote $SiO_2$ enrichment in the lithic fraction by sequestration of Fe in the metallic fraction (as long as the system remains above the oxygen fugacity defined by $Si-SiO_2$), and (2) even precursor materials with abundant olivine will produce Si-rich melts under low-degree partial melting if melting occurs at a eutectic involving albitic plagioclase (akin to the R4 study[44]). Both of these mechanisms were used to explain the enigmatic origin of extensive boninitic magmatism on the planet Mercury[46]. Although mechanism 2 cannot be used to explain the origin of NWA 11119 given its alkali-depleted bulk composition, mechanism 1 may apply to NWA 11119 given that its bulk composition has FeO abundances that are intermediate between the melts produced in the EH4 and H experimental studies. This intermediate FeO abundance indicates that NWA 11119 could have been derived by partial melting of a chondritic precursor at an $fO_2$ that is intermediate between the $fO_2$ investigated in the H and EH4 studies (IW-1 and IW-5, respectively). In fact, if the FeO abundance of NWA 11119 is reflective of the FeO abundance of its source, it would imply an $fO_2$ of approximately IW-4, if its source was in equilibrium with Fe-rich metal (computed using the methods outlined in ref.[47]). Partial melting studies of chondritic materials between IW-1 and IW-5 are lacking, but future studies aimed at this range in $fO_2$ may provide important constraints into the origin of NWA 11119, the origin of other evolved and reduced asteroidal samples, and the origin of reduced planetary bodies such as Mercury[48]. NWA 11119 provides the first unambiguous evidence for evolved volcanism in the earliest stages of Solar System history, prior to the assembly of the terrestrial planets.

## Methods

**Sample material**. A 4.2 cm × 3.3 cm × 2.3 cm rock sample of NWA 11119 (Fig. 1) was used for this study which is on deposit at the Institute of Meteoritics (IOM) at the University of New Mexico (UNM). This sample weighs 23 g and was removed from the main mass (Supplementary Fig. 1) that was found in Mauritania in December 2016. The IOM deposit sample was internally scanned using XCT at Johnson Space Center (JSC). A ~5 mg fragment was removed from the deposit sample for oxygen isotope studies at UNM. A ~200 mg fragment was removed from the deposit sample for X-ray diffraction studies at UNM. A ~500 mg fragment was removed from the deposit sample for trace element and Al–Mg dating studies at Arizona State University (ASU). A 0.9 cm × 0.55 cm thick section was cut off with a diamond saw and placed into a 1-inch epoxy mount for electron microscopy techniques at UNM and JSC.

**X-ray computed tomography**. XCT was conducted on the IOM deposit sample using a Nikon XTH 320, enclosed cabinet, micro-focus type CT scanner in the Astromaterials Research and Exploration Science (ARES) E-beam facility at JSC to obtain estimates on mineral modal abundances and vesicle content. The scan was obtained using a 175 kV accelerating voltage and 86 μA current. The deposit sample was enclosed in bubble wrap and placed inside a cylindrical cardboard dispenser to minimize any movement during the ~12 h scan. A 0.5 mm Cu filter was placed in front of the output window of the X-ray source during the scan. The scan consisted of 3141 projections with 16 frames per projection, which ensured a high-resolution and effective pixel size of 23.87 μm. Modal abundances and porosity estimates were calculated using the XCT processing program myVGL 3.0. The

vesicles/cavities in the deposit sample can be viewed in Supplementary Movie 1, and the mineralogy can be viewed in Supplementary Movie 2.

**Scanning electron microscopy**. Scanning electron microscopy (SEM) techniques were conducted on a JEOL 7600F SEM in the ARES/JSC on the slice of NWA 11119. This method was used to identify minerals and obtain images and X-ray maps of the section. The SEM is equipped with an energy dispersive spectrometer (EDS), which allowed for semi-quantitative measurements of phases. Backscattered electron (BSE) images (Supplementary Fig. 2) were acquired with a 15 kV accelerating voltage and 3.89 nA beam current. BSE and X-ray maps were acquired with a 15 kV accelerating voltage and 30 nA beam current. Mineral modal abundances were calculated by pixel counting using the image processing program ImageJ[49].

SEM techniques were also conducted on a Tescan Vega3 SEM equipped with a LaB6 gun at the IOM (UNM) on the slice of NWA 11119. This method was used to obtain quantitative data on the average groundmass. Conditions used were a 15 kV accelerating voltage and 1 nA beam current. Data on the matrix were obtained by drawing a free-hand area of interest in the shape of a square, which was then divided into four equal quadrants. The beam was moved in a raster pattern over each quadrant to collect an EDS spectrum, and the acquired data were then averaged for each section. Analyses on diopside and labradorite standards were also taken to test EDS reproducibility, and $SiO_2$ values were within 3% error of standard values. Density corrections were not applied to this data set. SEM matrix compositions are given in Supplementary Table 3.

**Electron probe microanalysis**. Electron probe microanalysis (EPMA) techniques were conducted using a JEOL 8530F hyperprobe at ARES/JSC on the slice of NWA 11119. This method was used to obtain quantitative chemical analyses on the phenocrysts and matrix grains. The hyperprobe is equipped with five wavelength-dispersive X-ray spectrometers (WDS) and an EDS system to obtain quantitative chemical analyses on phases. Silicates were analyzed using a 20 nA beam current, 15 kV accelerating voltage, and 1–5 μm spot size. Standards and elements were used as followed: diopside for Si, Ca, rutile for Ti, apatite-wilberforce for P, rhodonite for Mn, orthoclase for K, albite for Na, Sitkin anorthite for Ca, Si, and Al, San Carlos olivine for Mg, Springwater olivine for Si and Fe, chromite for Cr, Rockport fayalite for Fe, and Ni metal for Ni. Count times were 30 s (±15 s background) for each element. Sulfides and oxides were analyzed using a 30 nA beam current, 15 kV accelerating voltage, and 1–5 μm spot size. Standards and elements were used as followed: rutile for Ti, rhodonite for Mn, troilite for Fe, chromite for Cr, ilmenite for Fe and Ti, Ni metal for Ni, Co metal for Co, Zn metal for Zn, V metal for V, Cu metal for Cu, diopside for Ca, zircon for Zr, San Carlos olivine for Mg and Si, and Kakanui hornblende for Al. Count times were 30 s (±15 s background) for each element. Totals below or above 98–102 wt.% were excluded.

EPMA techniques were also conducted using a JEOL 8200 Superprobe at the IOM (UNM) on the slice of NWA 11119. Six sections of the matrix were mapped using quantitative X-ray mapping techniques with the Probe for Windows software. Each map was 300 × 300 pixels in size and acquired in two passes. Conditions used were a 15 kV accelerating voltage, 30 nA beam current, and 1 μm spot size. Standards and elements were used as followed: diopside for Si and Ca, olivine for Mg, albite for Na, spessartine for Mn, alumina for Al, pyrite for S, hematite for Fe, rutile for Ti, and chromite for Cr. Labradorite and ilmenite were included as secondary standards. The quantitative maps were processed using a custom MatLab script that uses element concentrations obtained through the six EPMA maps to identify the minerals in each frame. Totals below or above 90–105 wt.% were excluded. The minerals were then assigned a density to calculate a bulk density-corrected composition. Mineral densities for the matrix phases were assigned based on the average compositions of each phase as follows: plagioclase 2.73 $g/cm^3$, silica 2.28 $g/cm^3$, low-Ca pyroxene 3.62 $g/cm^3$, high-Ca pyroxene 3.44 $g/cm^3$, and opaques 4.85 $g/cm^3$. The average density for the matrix material was determined to be 3.12 $g/cm^3$, which was used to compute the bulk rock composition for NWA 11119. EPMA-calculated matrix compositions with and without density corrections are given in Supplementary Table 3. The XCT results were used to determine the mode of NWA 11119, which was also used to compute the bulk composition of NWA 11119. The XCT results indicated that the lithic portion of NWA 11119 consists of 89% phenocrysts and 11% matrix. The phenocrysts comprise 56% plagioclase, 30% tridymite, and 14% pyroxene. Mineral densities for the phenocrysts were assigned based on the average compositions of each phase as follows: plagioclase 2.74 $g/cm^3$, silica 2.28 $g/cm^3$, and pyroxene 3.28 $g/cm^3$. The density-corrected bulk composition for NWA 11119 is provided in Supplementary Table 3.

**X-ray diffraction**. X-ray diffraction (XRD) techniques were conducted on a Rigaku Smartlab X-ray diffractometer at UNM to determine the composition of the silica polymorph in NWA 11119. The diffractometer is equipped with a Cu-tube $K_\alpha$ source and a D/teX detector. ~200 mg of the sample was ground and homogenized into a whole-rock powder with a mortar and pestle, and then loaded into the diffractometer. The sample was scanned continuously with a range of 5°–140°, a step size of 0.02°, at a rate of 6°/min. Whole powder peak fitting was completed using MDI Jade software. XRD pattern for the bulk rock is exhibited in Supplementary Figure 5.

**Laser fluorination techniques**. Oxygen isotopes were obtained from four acid-washed fragments (0.9–1.5 mg) of NWA 11119 in the Center for Stable Isotopes at UNM using a laser fluorination technique. A fraction of NWA 11119 was crushed with a mortar and pestle, and four fractions were separated and washed with dilute HCl (6 M), high-purity alcohol, and distilled water, and then dried in a 60 °C oven. Molecular $O_2$ was obtained in a $BrF_5$-atmosphere, and isotope ratios were measured on a Delta PlusXL mass spectrometer, as described in ref.[50]. Isotopic measurements were obtained for $\delta^{17}O = (\delta^{17}O/\delta^{16}O)_{sample}/(\delta^{17}O/\delta^{16}O)_{SMOW}-1) \times 1000$, $\delta^{18}O = (\delta^{18}O/\delta^{16}O)_{sample}/(\delta^{18}O/\delta^{16}O)_{SMOW}-1) \times 1000$, and $\Delta^{17}O = \delta^{17}O - 0.528 \times \delta^{18}O$. San Carlos olivine was used for standardization. Oxygen isotope values are listed in Supplementary Table 6.

**Inductively coupled plasma mass spectrometry**. The Al–Mg isotope analyses were conducted in a clean laboratory in the Isotope Cosmochemistry and Geochronology Lab (ICGL) at ASU. A ~50 mg internal (free of fusion crust) fragment was powdered to obtain a whole-rock (WR) fraction and another internal ~200 mg fragment was processed for mineral separation. After careful crushing, sieving, and density separation, one pyroxene separate (Px A), three plagioclase-rich separates (Plag A, B, and C), and one silica-rich separate ($SiO_2$) were handpicked. The mineral separates were digested on a hotplate using a 3:1 $HNO_3$:HF mixture. To ensure complete dissolution, the WR was digested in a high pressure Parr bomb. Aliquots (5%) of the dissolved WR fraction and mineral separates were reserved for determining elemental abundances while the remainder of each solution was processed for Mg purification using methods described in ref.[51]. Excesses in $^{26}Mg$ from the decay of $^{26}Al$ are reported as $\mu^{26}Mg^*$, defined as the non-mass-dependent excesses in the $^{26}Mg/^{24}Mg$ ratio relative to the terrestrial (DSM3) standard in parts per million (ppm) notation. The Al/Mg ratios and Mg isotopes were measured using methods similar to those described in ref.[51]. We have since improved the precision of our Mg isotope analyses for samples with Mg ≥1 µg; specifically, two times the standard deviation on $\mu^{26}Mg^*$ for repeat runs of terrestrial synthetic and rock standards over the course of this study (i.e., 2 SD external reproducibility) was ±6 ppm[52]. Uncertainties in $\mu^{26}Mg^*$ are reported as either two times the standard error (2SE) from repeat runs of the sample ($n = 10$) or the 2 SD external reproducibility (±6 ppm; 54), whichever is larger. The slope and intercept of the $^{26}Al$–$^{26}Mg$ internal isochron were calculated using ISOPLOT[53]. The Al–Mg isotopic data are reported in Supplementary Table 7.

The major and rare earth element (REE) abundances (Supplementary Table 5) were measured in the Keck Laboratory at ASU using a Thermo iCAP-Q ICPMS. A fraction of the reserved 5% aliquot of each mineral separate and WR fraction was diluted in 3% $HNO_3$ and run in kinetic energy discrimination mode. Internal standards are prepared in house and run along with the samples. Additionally, terrestrial synthetic and rock standards were run after every five samples to monitor the instrument performance. Based on the reproducibility of these standards, we estimate the uncertainties on the major element and REE concentrations to be ~±3% and ~±10%, respectively.

**Data availability**. The authors declare that the data supporting the findings of this study are available within the paper and its Supplementary Information Files.

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

## Acknowledgements

We thank A. Aithiba for lending the deposit sample of NWA 11119 to the Institute of Meteoritics, which has enabled this research. We are thankful to the Maine Mineral and Gem Museum for the high-resolution photograph of the NWA 11119 main mass specimen. We thank M. Spilde, J. Lewis, B. Ha, and E. J. Peterson at the University of New Mexico, and E. Berger and D. K. Ross at ARES Johnson Space Center for technical assistance and support. We also thank A. Ramsey from Nikon for assistance processing the XCT videos. We are grateful to R. Hines, V. Rai, and S. Romaniello at Arizona State University for assistance in the Isotope Cosmochemistry and Geochronology Laboratory. The ICPMS analyses were supported by NASA Emerging Worlds grant #NNX15AH41G to M.W. F.M.M. acknowledges support for this research that was provided by NASA's Planetary Science Research Program.

## Author contributions

P.S. carried out all SEM and electron microprobe techniques. C.B.A. and F.M.M. assisted in data acquisition and interpretation. D.R.D. and M.W. conducted all ICPMS analyses, and assisted with the interpretation of these data. D.C. and R.Z. carried out all XCT scans and associated data processing. K.Z. conducted oxygen isotope techniques. All authors contributed to the writing of this manuscript.

## Additional information

**Competing interests:** The authors declare no competing interests.

