## [Peer Review File · Nature Communications]

Reviewers' comments:

Reviewer #1 (Remarks to the Author):

This paper is a very good work. Some of the data obtained by the authors are just superb. This meteorite is unique and the publication of this paper in Nature Comm. is justified. I strongly suggest the publication of this work in Nature Communications. I have only a few comments, most of them are minor. A few references are lacking.

- have you try to analyse (EMPA) the composition of the fusion crust? (and comparison with the WR composition) It is just a question, not a comment.

-line 35, "tertiary crust"? see below.

-line 44, after "Solar System", please give some references...

-line 59, indicate clearly the size of the crystals.

-lines 67-72, from "in comparison" to "21% silica". These three sentences are useless. When you see the picture of the hand specimen, it is obvious that a small surface cannot be representative of the whole rock. You have determined correctly the modal composition with the XCT instrument. Perfect! Please, do not waste space with useless discussion. Delete these 3 sentences.

-lines 87-89. Fig. S5 is difficult to read. I am surprised by the occurrence of cristobalite. Do you know where this phase is located. Do you have Raman data on the silica phenocrysts (they all look like tridymite? (not mandatory). Correct the sentences by: "... the silica phases include cristobalite and tridymite that formed at high temperature (references 4 and 5 are useless), but not quartz. The silica phenocrysts appear...". The P and T conditions are useless here (it is a volcanic rock – it is obvious that it cristallized at low P and high T, and certainly much below 1600°C).

-line 100. Give the range or the exact total alkali abundances and SiO₂ abundances. Delete "relative to typical achondrites". This rock is not a typical achondrite, if you want to do a comparison, compare it here with GRA or ALMA.

-Line 120. It is extremely frustrating to not have some major element abundances of the fractions, and other trace elements. Is it possible to get additional data in the revised version of the paper? These data are necessary to evaluate the modal composition of the WR sample.

- Line 128, please correct: "... (12-14), ALM-A an ureilitic trachyandesite (32), and the basaltic ungrouped achondrite NWA 7325 (15-16, 33)...".

-Lines 159-163. I agree with this para, but I think that a slight revision is necessary. Just as one swallow doesn't make a summer, neither does a single piece of SiO₂-rich meteorite demonstrates that the crust of its parent body is andesitic, dacitic or granitic. NWA 11119 is perhaps not representative of the crust of its parent body. We have rare lunar granites and the lunar crust is not granitic. We have rare andesitic or granitic debris in howardites and the crust of Vesta is certainly mafic. Even on Earth, plagiogranites are well known in oceanic crust which is gabbroic-basaltic... In the case of the ureilite parent body, we have two arguments that suggest a SiO₂-rich crust: first, we have ALM-A and some debris in polymict ureilites, and secondly ureilites are mantle restites whose trace element abundances suggest that trachyandesitic melts were extracted (Barrat et al., GCA 2016, 194, 163-178)..

-lines 182-191. The two hypotheses are not correctly discussed. ALM-A originated from the UPB, not NWA 7325 (see ref. 16 and 33). The possibility that NWA 7325 and 11119 originated from the same body should be more extensively discussed. These rocks are not necessary genetically linked. Notice that the material that was parental to NWA 7325 must have been older than 4563, and possibly formed 4566 Ma ago (ref. 33), like NWA 11119... Interestingly NWA 7325 displays a striking negative Tm anomaly ($Tm/Tm^*=0.86$, Barrat et al. GCA 2016, 176, 1-17, see table 3). Similarly, the REE pattern obtained in this work for NWA 11119 shows a pronounced negative Tm anomaly too ($Tm/Tm^*=0.91$), lower than achondrites (ureilites, eucrites, angrites), lunar, terrestrial and martian rocks...

-lines 192-225 (Discussion, last para)

* please include in the discussion the following work: Usui et al. (2015), MAPS 50, 759-781. Rewrite this section accordingly.

* The possibility that andesitic melts could be generated by melting of chondrites is rather recent in the literature. The number of experimental works on this topic is still limited and the REE abundances in the NWA 11119 WR (taken into account that the possible fraction of phenocrysts in the sample), is in the range of what we can expect in this case). The possibility that NWA 11119 represents a partial melt of an achondritic crustal rock is not convincing.

*lines 199-200. How do you know the size of the parent body (P of melting)? Please justify. Can you really exclude the possibility of a large planetary embryo?

-Figure 1 and Figure S1 are equivalent. Figure S1 is much better. I suggest to swap these figures.

-Figure 3C. Please use the NWA 7325 pattern obtained by Barrat et al. (2016 GCA- because Tm was measured) instead of the pattern from reference 33.

-Supplementary materials.

-EMPA (section 2.2). Can you indicate how you obtained the quenched melt/matrix compositions. What was the size of the spots?

-ICP-MS (section 2.6)

What are the accuracies for the REE ratios (including Tm/Tm* and Eu/Eu*)?

- Figure S2. What are the inclusions in plagioclase? Could you provide a larger BSE image for the quenched melt?

-Tables. Units are lacking (wt%, ppm)

-Tables, phase compositions. Please can you prepare a table with average compositions, cores, rims, range of compositions for the various phases, and provide a supplementary excel table for all the EMP analyses.

These comments should not be overemphasized. I like this work, and I think it is very good.

Jean-Alix Barrat

Reviewer #2 (Remarks to the Author):

NCOMMS-18-04835 Review by Philipp R Heck:

The authors report their finding of the earliest silicic volcanism in the solar system. They base their finding on a detailed petrologic and geochemical study of the rare achondritic meteorite NWA 11119 and its relative age based on Al-Mg systematics.

I anticipate that their results and conclusions are of great interest to the scientific community. Evidence for early solar system volcanism came predominantly from mafic rocks, what is presented here is new. It was not expected that silicic volcanism would occur that early. This finding now requires that silicic volcanism needs to be explained by early differentiation and asteroid evolution models that also need to explain volcanism that produced the more common mafic asteroidal rocks. In addition, this rock is of interest because of its unusual and currently unique combination of mineralogy, chemistry, and oxygen isotopic composition.

The authors' meticulous work is convincingly presented, very well written and accompanied with excellent illustrations and documentation.

The authors provide a thorough petrologic description of the rock. They combine state-of-the-art electron microscopy X-ray spectroscopy techniques, with X-ray tomography and X-ray diffraction to describe the texture, mineral chemistry and modal abundances of the rock. The rock clearly shows characteristics of an extrusive, silicic volcanic rock.

In addition to this and the bulk composition, the authors used laser fluorination mass spectrometry to measure three oxygen isotopes, a standard technique to classify meteorites, but particularly important for unusual achondrites such as this one. To obtain the key chronological information that makes their finding so important they measure Mg isotopes using state-of-the-art mass

spectrometry. All these analyses were necessary to properly compare this rock to other meteorites – a requirement to arrive at the presented conclusions.

The manuscript explains that silicic volcanism that early was not expected and compares the possible asteroidal origin of the meteorite with other meteorites that have similar characteristics. In addition to the meteorites in the discussion it might be of interest to add a brief comparison to silicic inclusions found in IIE iron meteorites, even though the O isotopes and mineralogies are different. The authors might want to consider even further emphasizing the importance of the possible implications of their findings in the broader context of early asteroid evolution for the non-specialist reader.

The age of this unusual rock reveals that evolved magmas in the solar system formed much earlier than anticipated until now. This fact alone requires a change how planetary scientists think about the early evolution of volcanism in our solar system. I am sure that this paper will generate many new studies of this unusual but important rock, such as Cr-isotope systematics and the use of other chronometers. This rock provides unprecedented information about early volcanism in the solar system. It also helps to better understand the possible genetic relationships between the increasing number of petrogenetically and geochemically interesting, ungrouped achondrites. Although meteorites such as this one are rare today, they might have been more frequent in the past and provide a glimpse into earliest geological processes in the solar system. In my opinion this work clearly deserves to be published in a high impact journal.

Reviewer #3 (Remarks to the Author):

Major comment:

This is the first report of an unique achondritic meteorite NWA 11119 with the oldest ^{26}Al - ^{26}Mg age among any known differentiated meteorites. Silica rich nature of the meteorite along with the old age may suggest that the meteorite may represent the evolved crustal rock on ancient asteroid. This is interesting study and detailed description along with the high precision isotope analyses and chemical analyses are qualified for publication in Nature Communication in respect to novelty. The movie of 3D imaging of sample is very useful for those who studies texture of such unusual meteorite.

However, the way the authors interpreted data to conclude “evolved tertiary crust” is not very strong. Towards the end of the paper, it is discussed that NWA 11119 is different from expected melt compositions of chondritic starting materials. Therefore, “evolved tertiary crust formation” is suggested as an alternative. However, if the meteorite derived from evolved crust, you may not find unfractionated REE pattern shown in Fig. 3B. There are no discussions on this REE pattern. Therefore, it is not clear how the igneous differentiation occurred on the asteroidal body to produce this meteorite. While the timing of the meteorite formation is determined in this work, they did not attempt to construct time evolution of the parent asteroid to form silica-rich rock represented by NWA 11119.

In this paper, ^{26}Al - ^{26}Mg chronology data were reported as initial $^{26}\text{Al}/^{27}\text{Al}$ ratio of the sample and converted age in absolute time scale by using D’Orbigny angrite as age anchor. The relative age after CAI should be also reported by comparing initial $^{26}\text{Al}/^{27}\text{Al}$ ratio to that of canonical CAI value ($5.2\text{E}-5$). If you do this, the relative age would become 3.49 ± 0.05 Ma, not within 3 Ma after CAI. By using Pb-Pb age of CAI 4567.3 ± 0.2 Ma by (24) and converted age of 4564.8 ± 0.3 Ma, you would calculate relative age of 2.5 ± 0.4 Ma. The discrepancy is due to inconsistency among CAI data, not achondrite data. It is better to present both relative ages, especially the data are obtained using Al-Mg system.

Fe/Mn ratios of pyroxene was not used in the discussion, which is often used as distinguish objects

from common parent bodies. Such data may be available to NWA 7325 and Alma-A, then it would be clear if they are related to NWA 11119 or not.

Minor comments:

- PDF file of appendix for electron probe data are very difficult to see.

- It is clear from comparison between Al-Mg data of "plagioclase" and EPMA data of plagioclase that mineral separates are not pure phase. It has more Mg, probably due to inclusions that are obvious in some of the images. There is no problem for the interpretation of Al-Mg data even they are not pure mineral phase. However, REE data from these separate minerals do not represent REE abundance of each mineral phase. This should be mentioned and discussed.

L 64-72: There are different set of numbers acquired for modes of minerals and components. This sounds too technical. It is good have overall assessment of these values from multiple dataset.

Reviewers' comments:

Reviewer #1 (Remarks to the Author):

This paper is a very good work. Some of the data obtained by the authors are just superb. This meteorite is unique and the publication of this paper in Nature Comm. is justified. I strongly suggest the publication of this work in Nature Communications. I have only a few comments, most of them are minor. A few references are lacking.

- have you try to analyse (EMPA) the composition of the fusion crust? (and comparison with the WR composition) It is just a question, not a comment.

We have not tried to analyze the fusion crust by EPMA.

-line 35, “tertiary crust”? see below.

Addressed below.

-line 44, after “Solar System”, please give some references...

We added three references, including Day et al. (2009), Bischoff et al. (2014), and Hahn et al (2017).

-line 59, indicate clearly the size of the crystals.

The crystal sizes have now been added to the text.

-lines 67-72, from “in comparison” to “21% silica”. These three sentences are useless. When you see the picture of the hand specimen, it is obvious that a small surface cannot be representative of the whole rock. You have determined correctly the modal composition with the XCT instrument. Perfect! Please, do not waste space with useless discussion. Delete these 3 sentences.

These three sentences have been deleted, and we only discuss the mode determined by XCT.

-lines 87-89. Fig. S5 is difficult to read. I am surprised by the occurrence of cristobalite. Do you know where this phase is located. Do you have Raman data on the silica phenocrysts (they all look like tridymite? (not mandatory). Correct the sentences by: “... the silica phases include

cristobalite and tridymite that formed at high temperature (references 4 and 5 are useless), but not quartz. The silica phenocrysts appear...”. The P and T conditions are useless here (it is a volcanic rock – it is obvious that it crystallized at low P and high T, and certainly much below 1600°C).

Upon re-inspection of our XRD data, we discovered that the MDI Jade software had erroneously assigned one of the minor anorthite peaks as the primary cristobalite peak with which it overlaps at approximately $22^\circ 2\theta$. Consequently, tridymite is the only silica phase in NWA 11119 as suspected by the reviewer. We thank the reviewer for urging us to re-visit and correct this result.

-line 100. Give the range or the exact total alkali abundances and SiO₂ abundances. Delete “relative to typical achondrites”. This rock is not a typical achondrite, if you want to do a comparison, compare it here with GRA or ALMA.

We removed “relative to typical achondrites”.

-Line 120. It is extremely frustrating to not have some major element abundances of the fractions, and other trace elements. Is it possible to get additional data in the revised version of the paper? These data are necessary to evaluate the modal composition of the WR sample.

Major element abundances have now been added to Table S5 (in addition to the REE data).

- Line 128, please correct: “...(12-14), ALM-A an ureilitic trachyandesite (32), and the basaltic ungrouped achondrite NWA 7325 (15-16, 33)...”.

This was corrected as requested.

-Lines 159-163. I agree with this para, but I think that a slight revision is necessary. Just as one swallow doesn't make a summer, neither does a single piece of SiO₂-rich meteorite demonstrates that the crust of its parent body is andesitic, dacitic or granitic. NWA 11119 is perhaps not representative of the crust of its parent body. We have rare lunar granites and the lunar crust is not granitic. We have rare andesitic or granitic debris in howardites and the crust of Vesta is certainly mafic. Even on Earth, plagiogranites are well known in oceanic crust which is gabbroic-basaltic... In the case of the ureilite parent body, we have two arguments that suggest a SiO₂-rich crust: first, we have ALM-A and some debris in polymict ureilites, and secondly ureilites are mantle restites whose trace element abundances suggest that trachyandesitic melts were extracted (Barrat et al., GCA 2016, 194, 163-178)...

We have revised this section slightly so that we no longer make the leap that NWA 11119 is representative of the bulk crust of a planetary body. We have modified the paragraph in the following way: “The bulk composition of NWA 11119, along with its early formation age as indicated by its ²⁶Al-²⁶Mg internal isochron, demonstrates unambiguously that andesitic-dacitic ~~crusts formed~~ **volcanism occurred** on a differentiated parent body within the first 3 Ma of Solar System history, extending both the age and degree of chemical evolution of planetary crustal materials on differentiated bodies in our Solar System.”

-lines 182-191. The two hypotheses are not correctly discussed. ALM-A originated from the UPB, not NWA 7325 (see ref. 16 and 33). The possibility that NWA 7325 and 11119 originated from the same body should be more extensively discussed. These rocks are not necessarily genetically linked. Notice that the material that was parental to NWA 7325 must have been older than 4563, and possibly formed 4566 Ma ago (ref. 33), like NWA 11119... Interestingly NWA 7325 displays a striking negative Tm anomaly ($Tm/Tm^*=0.86$, Barrat et al. GCA 2016, 176, 1-17, see table 3). Similarly, the REE pattern obtained in this work for NWA 11119 shows a pronounced negative Tm anomaly too ($Tm/Tm^*=0.91$), lower than achondrites (ureilites, eucrites, angrites), lunar, terrestrial and martian rocks...

We conducted a comparison of Fe/Mn ratios of pyroxenes between NWA 11119, ALM-A, and NWA 7325. It was clear from these data that NWA 11119 may be from a distinct parent body. We have modified the text to remove the two scenarios and conclude that NWA 11119 could originate from a previously unsampled parent body. The Fe/Mn plot is now in Figure S5. The one aspect that we do not agree on is the presence of the Tm anomaly in the NWA 11119 whole-rock (WR). As mentioned by this reviewer, we can calculate a Tm/Tm^* value of ~ 0.9 ; however, when the error on this value is considered (15%), we cannot definitively state that there is a Tm anomaly in the WR REE pattern for NWA 11119.

-lines 192-225 (Discussion, last para)

* please include in the discussion the following work: Usui et al. (2015), MAPS 50, 759-781. Rewrite this section accordingly.

This works have been incorporated and the discussion modified accordingly. The Usui results are particularly important and relevant here as they allow us to develop a testable hypothesis about the origin of NWA 11119 through partial melting of chondrite under moderately reducing conditions. The re-worked discussion removes the portion of the discussion that 2 of the 3 reviewers did not like about tertiary crust production.

* The possibility that andesitic melts could be generated by melting of chondrites is rather recent in the literature. The number of experimental works on this topic is still limited and the REE abundances in the NWA 11119 WR (taken into account that the possible fraction of phenocrysts in the sample), is in the range of what we can expect in this case). The possibility that NWA 11119 represents a partial melt of an achondritic crustal rock is not convincing.

We have modified the discussion so that it does not argue in favor of NWA 11119 representing tertiary crust, and we provide a more thorough discussion on the implications of the trace element abundances and bulk composition of NWA 11119.

*lines 199-200. How do you know the size of the parent body (P of melting)? Please justify. Can you really exclude the possibility of a large planetary embryo?

We do not know the pressure, and no, a large body cannot be excluded. We deleted the text “at conditions relevant to small-bodied asteroids”.

-Figure 1 and Figure S1 are equivalent. Figure S1 is much better. I suggest to swap these figures.

The swap was made.

-Figure 3C. Please use the NWA 7325 pattern obtained by Barrat et al. (2016 GCA- because Tm was measured) instead the pattern from reference 33.

The REE pattern from Barrat et al. (2016) is now used in Fig. 3C.

-Supplementary materials.

-EMPA (section 2.2). Can you indicate how you obtained the quenched melt/matrix compositions. What was the size of the spots?

We have added additional information regarding the quenched melt compositions. For the record, A 1 μm beam was used for the maps, which has been added to this section.

-ICP-MS (section 2.6)

What are the accuracies for the REE ratios (including Tm/Tm* and Eu/Eu*)?

In section 2.6, we have now specified that our analytical uncertainty for the REE concentrations is $\sim\pm 10\%$ (while for major elements it is $\sim\pm 3\%$). With this information, it is straightforward to propagate these errors to estimate errors on elemental ratios ($\sim\pm 15\%$ for trace element ratios and $\sim\pm 5\%$ for major element ratios, after rounding up).

- Figure S2. What are the inclusions in plagioclase? Could you provide a larger BSE image for the quenched melt?

The inclusions are quenched material. Two images have been added to Figure S2 of the quenched melt and inclusions.

-Tables. Units are lacking (wt%, ppm)

Units are now given in the tables.

-Tables, phase compositions. Please can you prepare a table with average compositions, cores, rims, range of compositions for the various phases, and provide a supplementary excel table for all the EMP analyses.

We have added a table (Table 1) to the main text of the manuscript with average compositions, cores, rims, range of compositions for the various phases. An Excel file with all EPMA analyses is included in the supplementary tables. All supplementary data tables exist in a single MS excel file with each tab representing a different Supplementary Table.

These comments should not be overemphasized. I like this work, and I think it is very good.

Jean-Alix Barrat

Reviewer #2 (Remarks to the Author):

NCOMMS-18-04835 Review by Philipp R Heck:

The authors report their finding of the earliest silicic volcanism in the solar system. They base their finding on a detailed petrologic and geochemical study of the rare achondritic meteorite NWA 11119 and its relative age based on Al-Mg systematics.

I anticipate that their results and conclusions are of great interest to the scientific community. Evidence for early solar system volcanism came predominantly from mafic rocks, what is presented here is new. It was not expected that silicic volcanism would occur that early. This finding now requires that silicic volcanism needs to be explained by early differentiation and asteroid evolution models that also need to explain volcanism that produced the more common mafic asteroidal rocks. In addition, this rock is of interest because of its unusual and currently unique combination of mineralogy, chemistry, and oxygen isotopic composition.

The authors' meticulous work is convincingly presented, very well written and accompanied with excellent illustrations and documentation.

The authors provide a thorough petrologic description of the rock. They combine state-of-the-art electron microscopy X-ray spectroscopy techniques, with X-ray tomography and X-ray diffraction to describe the texture, mineral chemistry and modal abundances of the rock. The rock clearly shows characteristics of an extrusive, silicic volcanic rock.

In addition to this and the bulk composition, the authors used laser fluorination mass spectrometry to measure three oxygen isotopes, a standard technique to classify meteorites, but particularly important for unusual achondrites such as this one. To obtain the key chronological information that makes their finding so important they measure Mg isotopes using state-of-the-art mass spectrometry. All these analyses were necessary to properly compare this rock to other meteorites – a requirement to arrive at the presented conclusions.

The manuscript explains that silicic volcanism that early was not expected and compares the possible asteroidal origin of the meteorite with other meteorites that have similar characteristics. In addition to the meteorites in the discussion it might be of interest to add a brief comparison to silicic inclusions found in IIE iron meteorites, even though the O isotopes and mineralogies are different. The authors might want to consider even further emphasizing the importance of the possible implications of their findings in the broader context of early asteroid evolution for the non-specialist reader.

We have modified this section to remove many of the specific scenarios and only focus on whether NWA 11119 is linked to existing samples or is from a distinct parent body. It was clear from the Fe/Mn data that NWA 11119 is likely from a distinct parent body. We have expanded upon the

discussion of how such an anomalous asteroid could have formed through melting of chondritic material, and we even now propose a testable hypothesis for the origin of NWA 11119, which should meet the sentiment of this comment to put this single exciting meteorite into a broader context.

The age of this unusual rock reveals that evolved magmas in the solar system formed much earlier than anticipated until now. This fact alone requires a change how planetary scientists think about the early evolution of volcanism in our solar system. I am sure that this paper will generate many new studies of this unusual but important rock, such as Cr-isotope systematics and the use of other chronometers. This rock provides unprecedented information about early volcanism in the solar system. It also helps to better understand the possible genetic relationships between the increasing number of petrogenetically and geochemically interesting, ungrouped achondrites. Although meteorites such as this one are rare today, they might have been more frequent in the past and provide a glimpse into earliest geological processes in the solar system. In my opinion this work clearly deserves to be published in a high impact journal.

Reviewer #3 (Remarks to the Author):

Major comment:

This is the first report of an unique achondritic meteorite NWA 11119 with the oldest ^{26}Al - ^{26}Mg age among any known differentiated meteorites. Silica rich nature of the meteorite along with the old age may suggest that the meteorite may represent the evolved crustal rock on ancient asteroid. This is interesting study and detailed description along with the high precision isotope analyses and chemical analyses are qualified for publication in Nature Communication in respect to novelty. The movie of 3D imaging of sample is very useful for those who studies texture of such unusual meteorite.

However, the way the authors interpreted data to conclude “evolved tertiary crust” is not very strong. Towards the end of the paper, it is discussed that NWA 11119 is different from expected melt compositions of chondritic starting materials. Therefore, “evolved tertiary crust formation” is suggested as an alternative. However, if the meteorite derived from evolved crust, you may not find unfractionated REE pattern shown in Fig. 3B. There are no discussions on this REE pattern. Therefore, it is not clear how the igneous differentiation occurred on the asteroidal body to produce this meteorite. While the timing of the meteorite formation is determined in this work, they did not attempt to construct time evolution of the parent asteroid to form silica-rich rock represented by NWA 11119.

We have softened on the idea that this meteorite represents a tertiary crust based on the suggestions of reviewer 1 and this reviewer. We now discuss the implications of the trace element abundances on the origin of NWA 11119 and consider alternative mechanisms to enable silica enrichment this early in Solar System history through chondrite partial melting processes.

In this paper, ^{26}Al - ^{26}Mg chronology data were reported as initial $^{26}\text{Al}/^{27}\text{Al}$ ratio of the sample

and converted age in absolute time scale by using D'Orbigny angrite as age anchor. The relative age after CAI should be also reported by comparing initial $^{26}\text{Al}/^{27}\text{Al}$ ratio to that of canonical CAI value ($5.2\text{E-}5$). If you do this, the relative age would become 3.49 ± 0.05 Ma, not within 3 Ma after CAI. By using Pb-Pb age of CAI 4567.3 ± 0.2 Ma by (24) and converted age of 4564.8 ± 0.3 Ma, you would calculate relative age of 2.5 ± 0.4 Ma. The discrepancy is due to inconsistency among CAI data, not achondrite data. It is better to present both relative ages, especially the data are obtained using Al-Mg system.

Text has been added to the relevant section indicating the Al-Mg ages obtained using different age anchors. As suggested by the reviewer, we now clarify that the formation time of NWA 11119 indicated by Al-Mg ages is 2.5-3.5 Ma after the formation of the first-formed refractory solids in the solar protoplanetary disk.

Fe/Mn ratios of pyroxene was not used in the discussion, which is often used as distinguish objects from common parent bodies. Such data may be available to NWA 7325 and Alma-A, then it would be clear if they are related to NWA 11119 or not.

We conducted a comparison of Fe/Mn ratios of pyroxenes between NWA 11119, ALM-A, and NWA 7325. It was clear from these data that NWA 11119 is likely from a distinct parent body. We have modified the text to remove the two scenarios and conclude that NWA 11119 likely originates from a previously unsampled parent body. The Fe/Mn plot is now in Figure S5.

Minor comments:

- PDF file of appendix for electron probe data are very difficult to see.

We have added a table (Table 1) to the main text of the manuscript with average compositions, cores, rims, range of compositions for the various phases. An Excel file with all EPMA analyses is included in the supplementary tables. All supplementary data tables exist in a single MS excel file with each tab representing a different Supplementary Table.

- It is clear from comparison between Al-Mg data of "plagioclase" and EPMA data of plagioclase that mineral separates are not pure phase. It has more Mg, probably due to inclusions that are obvious in some of the images. There is no problem for the interpretation of Al-Mg data even they are not pure mineral phase. However, REE data from these separate minerals do not represent REE abundance of each mineral phase. This should be mentioned and discussed.

Text has been added to the relevant section explaining that the mineral separates are not likely to be pure phases.

L 64-72: There are different set of numbers acquired for modes of minerals and components. This sounds too technical. It is good have overall assessment of these values from multiple dataset.

We have focused on the values attained from the XCT data because they are the most reliable, as suggested by the first reviewer.

References Cited:

- Bischoff, A. et al., 2014. Trachyandesitic volcanism in the early Solar System. *Proceedings of the National Academy of Sciences of the United States of America*, 111(35): 12689-12692.
- Day, J.M.D. et al., 2009. Early formation of evolved asteroidal crust. *Nature*, 457(7226): 179-U68.
- Hahn, T.M., Lunning, N.G., McSween, H.Y., Bodnar, R.J., Taylor, L.A., 2017. Dacite formation on Vesta: Partial melting of the eucritic crust. *Meteoritics & Planetary Science*, 52(6): 1173-1196.